# Optical Performance and Moisture Stability Enhancement of Flexible Luminescent Films Based on Quantum-Dot/Epoxy Composite Particles

**DOI:** 10.3390/nano11082100

**Published:** 2021-08-18

**Authors:** Guanwei Liang, Yong Tang, Jiarui Huang, Jiasheng Li, Yikai Yuan, Shu Yang, Zongtao Li

**Affiliations:** 1National and Local Joint Engineering Research Center of Semiconductor Display and VLC Devices, South China University of Technology, Guangzhou 510640, China; megwliang@yeah.net (G.L.); ytang@scut.edu.cn (Y.T.); 202021002916@mail.scut.edu.cn (J.H.); Jiasli@foxmail.com (J.L.); yangshme@scut.edu.cn (S.Y.); 2Guangdong Provincial Key Laboratory of Semiconductor Micro Display, Foshan Nationstar Optoelectronics Company Ltd., Foshan 528000, China; yuanyikai@nationstar.com

**Keywords:** quantum dots, composite particles, luminous efficiency, moisture stability

## Abstract

Quantum dots (QDs) have been widely applied in luminescent sources due to their strong optical characteristics. However, a moisture environment causes their quenching, leading to an inferior optical performance in commercial applications. In this study, based on the high moisture resistance of epoxy resin, a novel epoxy/QDs composite particle structure was proposed to solve this issue. Flexible luminescent films could be obtained by packaging composite particles in silicone resin, combining the hydrophobicity of epoxy resin and the flexibility of PDMS simultaneously. The photoluminescence and light extraction were improved due to the scattering properties of the structure of composite particles, which was caused by the refractive index mismatch between the epoxy and silicone resin. Compared to the QD/silicone film under similar lighting conditions, the proposed flexible film demonstrated increased light efficiency as well as high moisture stability. The results revealed that a light-emitting diode (LED) device using the composite particle flexible (CPF) structure obtained a 34.2% performance enhancement in luminous efficiency as well as a 32% improvement in color conversion efficiency compared to those of devices with QD/silicone film (QSF) structure. Furthermore, the CPF structure exhibited strong thermal and moisture stability against extreme ambient conditions of 85 °C and 85% relative humidity simultaneously. The normalized luminous flux degradation of devices embedded in CPF and QSF structures after aging for 118 h were ~20.2% and ~43.8%, respectively. The satisfactory performance of the CPF structure in terms of optical and moisture stability shows its great potential value in flexible commercial QD-based LED displays and lighting applications.

## 1. Introduction

Quantum dots (QDs) have been attracting great attention and have gradually become one of the most promising color-conversion materials in white light-emitting diodes (LEDs), because of their high quantum yield, high color rendering quality, and narrow emission angle [1,2,3,4,5,6]. They are widely used in LED encapsulation as photoluminescence materials for next-generation lighting and display devices [7,8,9]. Various breakthroughs regarding the luminous efficacy of QD-based LEDs have been achieved gradually, and the efficacy of QD-based LEDs has reached 200 lumens per watt [10]. In QD-based LED encapsulation, QDs are dissolved in polymer when fabricating composite materials to prevent them from moisture and oxidation [11]. Meanwhile, silicone is the most common packaging material because of its excellent flexibility, mechanical strength, optical transmittance, as well as thermal stability [12,13,14]; however, it suffers from bad moisture stability. The moisture and/or oxygen may lead to the aggregation, surface destruction, and photoluminescence quenching of QDs and thus, negatively affect the lifetime of QD-based LED devices [15,16]. To date, numerous packaging methods have been proposed to solve these issues, such as chemically passivating the QD surface during the chemical synthesis process [17,18]. Using polymer materials to protect the QDs is the most common QD encapsulation approach. For instance, CdS/PMMA core/shell nanoparticles are fabricated to maintain the optical properties of QDs while effectively protecting them from environmental perturbation [19]. Recently, a super-hydrophobic microstructure was proposed to enhance the water stability of QDs. Xuan et al., embedded QDs into a super-hydrophobic porous organic polymer framework [20]. QDs have been also incorporated into a porous matrix to isolate moisture [21,22]. Moreover, superhydrophobic nanosilica particles have been coated on the QDs/polymer films to prevent moisture from penetrating the films [23]. QD/polymer fibers prepared by electrospinning demonstrate a new and effective way to enhance QD moisture stability, which can be used to prepare water-resistant hybrid membranes [24,25]. Previous studies have indicated that QD encapsulation by a polymer matrix is the most efficient approach to enhance QD moisture stability. However, the above-mentioned methods may involve complex material selection and fabrication processes, which may negatively affect the optical properties of QDs. Furthermore, with regard to QD-LED packaging, limited studies have focused on device moisture stability, which has been mostly ignored to consider the optical performance. The addition of anti-moisture materials or structure could decrease the device’s lighting efficiency [23]. The main challenge is to realize a QD/polymer composite matrix that simultaneously exhibits light efficiency and moisture stability.

Epoxy resin is usually used in outdoor LED encapsulation owing to its high optical stability and moisture resistance [26,27], and always as the encapsulation resin for QDs [28]. However, the curing epoxy demonstrates a high degree of hardness, which limits its application in devices requiring flexibility, such as wearable devices. Nevertheless, the epoxy can be ground into micro particles to realize a QD/epoxy composite matrix, which can be easily dissolved in traditional packaging materials such as silicone resin. In addition, the diameter of QD is about several to a hundred nanometers [29], and its weak scattering ability may be compensated by the micro particles [30]. The existence of difference indices between the packaging materials is beneficial to enhance the light scattering [7,31], which contributes to improving the QD photoluminescence. Based on the fabrication of a QD/epoxy composite structure, the light path can be controlled by packaging the composite materials in the silicone resins. The composite particle structure can be an alternative method to enhance both the device’s moisture stability and light efficiency.

In this study, QDs and epoxy resin composite particles were fabricated to enhance the optical performance and moisture stability of flexible luminescent films. The fabrication and optical behaviors of the composite particle structure were thoroughly investigated. Finally, flexible QD films and QD-based LEDs with similar lighting effects were prepared and tested to confirm the effect of the composite particles. Compared to the traditional packaging structure obtained by dissolving QDs in silicone resin, the results obtained for the composite structure in this study illustrated that LED devices with the composite particle film (CPF) can greatly enhance the luminous and color-conversion efficiencies as well as moisture stability. Therefore, the CPF structure can be an effective and suitable candidate for QD powders for manufacturing flexible LED devices in the industry.

## 2. Experimental and Measurement

### 2.1. Composite Particles Preparation

In this study, CdSe/ZnS core-shell QDs with an emission peak of 570 nm were bought from Beida Jubang Co. Ltd (Beijing, China). as in our previous works [29], which quantum yield is about 85%. First, 120 mg of QDs was dissolved in chloroform, followed by the addition of 4 g of epoxy resin (IK0010, IK INABATA & Co., Ltd., Tokyo, Japan) into the mixture. Subsequently, in order to completely volatile the chloroform, the mixture was subjected to centrifugal stirring with 400 rpm revolution rate and 300 rpm rotation rate for 2 h using a mixer (FY-821, Shenzhen Fuyuanxiang Electronic Equipment Co., Ltd., Shenzhen, China). Subsequently, 4 g of epoxy crossing resin was added to the mixture, and the mixture was stirred in vacuum to ensure uniform mixing and removal of bubbles. Finally, the mixture was tape-casted on a PET film and cured at 150 °C for 3 h. The curing films were cut into small pieces and then ground at 450 rpm for 5 h by a ball grinder. This process is illustrated in detail in Figure 1. After filtering, uniform QDs/epoxy composite particles (QE CPs) were obtained; the photograph of the composite particles is shown in Figure 1d.

### 2.2. Fabrication of Flexible Luminescent Films

To obtain a flexible film, the composite particles were packaged in the polydimethylsiloxane (PDMS, Dow Corning Sylgard-184, Midland, MI, USA) silicone resin. It can easily achieve different lighting effects by varying the composite particle concentration from 0.09 to 0.37 wt%. The QD concentration is defined as the ratio of QD mass to the mass of resin. After uniformly mixing the composite particles and PDMS, the mixture was cast and cured at 110 °C for 0.5 h using a mold casting method. Under the same curing condition, pure PDMS and epoxy films were fabricated by a standard mold to study their mechanical properties. A photograph of the prepared composite particle flexible (CPF) films is shown in Figure 1e. For comparison, a QD/silicone (QS) film was also fabricated by directly dissolving QDs in pure PDMS silicone. The QD concentration of QS films was adjusted to achieve the same lighting effect with the CPF-structured films.

### 2.3. Characterizations and Measurements

The morphology of composite particles was characterized by field emission scanning electron microscopy (FE-SEM, Zeiss^®^ Merlin, Oberkohen, Germany). The diameter distribution of composite particles was measured using Image J software. Elemental mapping images of the composite particles were also analyzed by Zeiss SmartEDX (Oxford, UK) to study the element mass ratio distribution on the surface of composite particles. The high-resolution transmission electron microscope (HRTEM) images of CdSe/ZnS core-shell QDs in epoxy resin were measured by a field emission transmission electron microscopy (JEM-1400 PLUS, JEOL, Tokyo, Japan). The TEM samples were sectioned by an ultramicrotome (UC7/FC7, LEICA, Wetzlar, Germany) at room temperature, and used for measurement. The FTIR spectra were tested by a Fourier Transform infrared spectroscopy (Nicolet IS50, Thermo Fisher Scientific Inc., MA, USA). The stress-strain curves of the PDMS and epoxy films were obtained through testing with a tensile testing machine (WDW-01A, Jinan Fine Experimental Instrument Co., Ltd., Jinan, China). During the tensile test, the stretching speed was maintained at 2 mm/min. The diffuse transmittance and absorbance were measured using a dual-beam UV-Vis spectrophotometer (TU-1901, Beijing Persee General Instrument Co., Ltd., Beijing, China). Details of the measurement of asymmetric bidirectional transmittance distribution function (BTDF) can be found in our previous report [32], which evaluates the scattering ability of the films. The photoluminescence (PL) was tested by a fluorescence spectrophotometer (RF-6000, Shimadzu, Japan) with an excitation wavelength at 365 nm. The optical performance of the LED devices with QD films was tested on an integrating sphere system including a 0.5-m-diameter integrating sphere, a spectrometer (Otsuka LE5400, Otsuka Electronics Co., Ltd., Osaka, Japan), and a DC power supply (Keithley 2425, Tektronix, Inc., Beaverton, USA). Blue COB-LEDs with sixteen chips were used to excite the remote QD films for white lights. The optical performance of two types of structural films was tested under different driving currents varying from 50 to 1000 mA. For the aging process, the CPF and QS films were placed in a thermostatic and humidistatic machine with ambient conditions of 85 °C and 85% relative humidity. After aging for different times, two types of structural films were coated on the COB devices to investigate their optical performance.

## 3. Results and Discussions

To study the morphology of the composite particles, Figure 2a shows the SEM image of the composite particles after grinding. It can be observed that the composite particles with spherical morphology exhibit a good dispersion and present a uniform diameter distribution. A total of 250 pieces of composite particles were used to calculate for average diameters; the results are shown in Figure 2c. The composite particles exhibit a concentrated distribution with an average diameter of about 40 μm.

As shown in Figure 1d, the composite particles show bright yellow emission under 365 nm, indicating that an effective package of QDs and epoxy composite particles was built. The QD/epoxy composite matrix can be ground into particles easily. To characterize the QD distribution in the epoxy, the energy dispersive spectroscopy (EDS) of composite particles was performed; the results are shown in Figure 2b,d,e. From Figure 2b, it is seen that the composite particles produce a rough surface after grinding, which may cause a change in incident light transfer direction. The element content measurement results (Figure 2d) show that a small amount of QD element content was detected, indicating that CdSe/ZnS QDs seldom distribute on the epoxy surface. It can be inferred from the EDS mapping results that QDs are distributed in the inset of epoxy resin, which contributes to protecting QDs from water and oxygen. To further support this issue, Figure 2e shows the HRTEM images of QDs in epoxy resin. The measured QDs demonstrate uniform size distribution and QD are relatively evenly distributed in epoxy resin. There is no obvious QD aggregation in epoxy resin. Figure 2f shows the FTIR spectra of the QDs and composite particles. Both the QDs and composite particles commonly exhibit the bands at 2846 and 2920 cm^−1^, which are attributed to the presence of -CH_3_ and -CH_2_ groups from the QD aliphatic surface light and species [17]. Typically, the composite particles exhibited a strong band at 1736 cm^−1^ from the C=O stretching of the ester group [33]. It is also indicated that the QDs are distributed in the epoxy resin. The epoxy film exhibits a good moisture ability but strong rigidity, which is not desirable to fabricate luminescent films in flexible lighting and display devices. It can be seen from Figure 2g that PDMS film demonstrates great flexibility. The addition of QDs slightly influenced the flexibility of CPF films, as shown in the inset of Figure 2g. To further study the mechanical properties of PDMS and epoxy films, a tensile test was carried out; the process is shown in Figure 2h. Figure 2i illustrates the representative stress-strain curve. Significantly, the PDMS film performs elastic deformation against stress while the epoxy film exhibits a higher elastic modulus compared to that of the PDMS film. During the elastic deformation stage, based on the proportional relationship between the stress and strain, the elastic modulus can be easily calculated from the stress-strain curve. The calculated elastic modulus of PDMS was about 1.57 MPa, while the CPF films exhibited a similar value. However, the epoxy film exhibited strong rigidity with a large elastic modulus at about 376.15 MPa. During these procedures, the combination of the respective advantage of PDMS and epoxy resin is essential to be used in flexible devices.

To obtain film flexibility, the composite particles were encapsulated in silicone resin, and different composite particle concentration would change the film optical performance. The CPF films with different composite particle concentration were prepared to study these issues (Figure 3). The diffuse transmittance of CPF films related to composite particle concentration is illustrated in Figure 3a. As the QD concentration increased from 0.09 to 0.37 wt%, the diffuse transmittance decreased gradually. The CPF films exhibited lower transmittance at shorter wavelengths and higher transmittance at longer wavelengths due to the light absorbance of QDs. Particularly, the diffuse transmittance at 450 nm decreased from 53% to 13%, indicating a strong QD absorbance. Meanwhile, the addition of composite particles may change the light scattering performance caused by the index difference between the PDMS and epoxy resin. The BTDF normalized intensity was tested to evaluate the film scattering ability; the results are shown in Figure 3b. The normalized angular intensity distribution of the film was tested after passing a 632-nm laser beam through the testing films. There exists a significant peak at 0° in pure PDMS as well as CPF films with a relatively low QD concentration of 0.09 wt%. As the composite particle concentration increased, more light scatters to larger angles due to the increase in the scattering ability. When the QD concentration is larger than 0.31 wt%, the scattering ability becomes saturated and the intensity begins to decrease because of the lower transmittance and light absorbance. It is evident that the composite particles can enhance the light scattering path, which may contribute to improving the QD excitation. Figure 3c shows the absorbance and PL intensity of CPF films. The CPF films exhibit an absorbance peak at about 552 nm. The composite particle concentration affects the absorbance to a large degree, and a larger concentration leads to greater light absorbance. Subsequently, the PL intensity with an emission peak located at 567 nm becomes stronger as the composite particle concentration increased, and it finally reaches a balance point, which is limited by the film transmittance. The PL intensity profile reveals that more QD light could be obtained by increasing the composite particle concentration.

It is well-known that the lighting effect can be easily adjusted by controlling the phosphor concentration. CPF films with various composite particle concentrations demonstrate different lighting performances. This behavior warrants further investigation. Figure 3d illustrates the optical performance of the blue COB-LED devices under excitation. By controlling the composite particle concentration, different lighting effects with various CCTs, as well as different luminous intensities, could be obtained. The luminous flux first increased and then decreased as the QD concentration increased. This is because more blue light was converted to yellow light, to which the luminous flux is most sensitive. Moreover, the light extraction could be affected by the larger concentration of the composite particles, which decreases the film transmittance. The highest luminous flux was 120.8 lm with a CCT of 6757 K when the QD concentration was 0.25 wt%. The change in radiant power can explain the conversion, which decreases gradually as the QD concentration increased. Thus, more blue light would be absorbed by QDs and converted to QD light. Therefore, CPF films enable obtaining different lighting effects using CPF films.

To ensure the practical application potential of composite particles, a QSF structure was fabricated by direct dissolving of QD in silicone, and its performance was compared with that of the CPF structure. Both structures were adjusted the QD concentration to ensure the same lighting effect. As can be observed in Figure 4a, COB-LED-based devices with both these structures exhibit the same lighting effect with a CCT of about 4761 K at driving current of 150 mA. The inset shows the structure of COB-LED device. The QD films were coated on blue COB-LED devices. Under a similar CCT, the devices with a CPF structure demonstrated better luminous efficiency (LE) than those of devices with a QSF structure. The LE for devices with a CPF structure was 73.4 lm/W, a 34.2% enhancement compared to those of devices with QSF. This can also be seen from the spectra of the two structures in Figure 4a. The intensity of the CPF structure is higher than that of QSF at both the blue and QD light wavelengths. To analyze the LE improvement clearly, the radiant power of the unexcited blue light, as well as QD light, were calculated by integrating the spectrum intensity from 380 to 500 nm wavelengths for blue light and 500 to 780 nm wavelengths for QD light; the results are illustrated in Figure 4b. For the unexcited blue light, the QSF structure exhibited a radiant power of 86.36 mW, which was lower than that of the CPF structure (95.52 mW). The CPF structure contributed positively to blue light extraction, resulting in a 10.6% enhancement of blue light radiant power. For remote QD films, a great interface loss would occur because of the total internal reflectance (TIR) between the film and air. Due to scattering from the composite particles, the TIR loss would be weakened, and the light extraction would be facilitated. The QD light radiant power of the CPF structure is 198.98 mW, which corresponds to a 30.2% improvement than that of the QSF structure. This could be due to the spatial distribution of QDs and the utilization improvement of incident blue light. Moreover, the path length of blue light is increased due to scattering effect, thus enhancing the blue light excitation. Compared to the QSF structure, the CPF structure possesses more blue light and QD light. It can be concluded that the CPF structure contributes positively to improve the color conversion efficiency (CCE). The CCE is defined as the ratio between the emission radiant power from the QDs to the absorbed blue light radiant power, which determines the output color coordinates of LEDs. As shown in Figure 4b, the CCE of the CPF structure is 29.8%, which is 32% higher than that of QSF; this indicates that the energy transfer of the CPF structure is more efficient than that of QSF. The CPF structure exhibits a better optical performance than the QSF structure, which proves that the proposed CPF structure is more suitable for illumination and display.

To better understand the optical enhancement, the light transfer schematic diagram is shown in Figure 4c. The different optical performances of two structure types can be explained as follows. With regard to the blue light power, due to the TIR loss between the air and silicone films, a large amount of blue light would be backscattered and re-absorbed by QD and package material. Based on the above discussion, it can be stated that the scattering ability of the CPF structure is stronger than that of the QSF structure. To further understand the scattering effect for these two kinds of structure, their BTDFs under similar lighting effects are shown in the inset. The lighting intensity of the CPF structure is more uniform than that of QSF, and the lighting angle is wider. The blue light could be scattered in different directions, thus increasing the probability of light escapes and decreasing the TIR loss. Furthermore, the scattering effect is an important feature that controls the excitation of QD photoluminescence. The blue light path length is greatly increased, which contributes to exciting more QDs. However, there exists an index mismatch between the epoxy (~1.50) and silicone resin (~1.41); therefore, the incident blue light is trapped in the composite. This trapping increases the blue light absorption by QDs, leading to more QD light production. Similarly, the scattering effect may also positively affect the QD light extraction. Thus, the CPF structure is beneficial to improve the light output.

To investigate the film optical stability under different driving currents varying from 50 to 1000 mA, the luminous flux and luminous efficiency of LED devices with two kinds of structural films were studied in Figure 5a. Although the luminous flux of both structural films increases, the luminous efficiency decreases as the driving current increased. The CPF structure performs better than the QSF structure, and the improvement becomes greater as the driving current increased. At a driving current of 50 mA, the luminous flux of the CPF structure is 40.3 lm while that of the QSF structure is 29.5 lm, indicating an enhancement of 36.6% in the former. When the current increased to 1000 mA, the luminous flux of the CPF structure is 567.6 lm, which is 58.9% greater than that of the QSF structure. The luminous efficiency also demonstrates the similar tendency as the luminous flux changed. These results reveal that the CPF structure demonstrates better optical stability than the QSF structure.

These results demonstrate the enhancement of light efficiency of LEDs with different packaging structures. To study the film thermal and moisture stability, two types of structural films were placed in the ambient conditions of 85 °C and 85% RH and with different aging times. The normalized intensity of the two types of structural films is shown in Figure 5b, which indicates the moisture effect on the photoluminescence quenching of QDs. The luminous flux decreases sharply and becomes smoother as the aging time increased, indicating that the QDs were affected by moisture severely. A comparison of the aging curves of two types of structural films reveals that the luminous flux degradation of the CPF structure is much smaller than that of QSF. After aging for 118 h, the CPF structure could maintain a 79.8% luminous flux, but the QSF structure exhibited a luminous flux of only 56.2%. These results illustrate that the moisture stability of the QDs is significantly improved by the CPF structure. This could be attributed to the anti-moisture ability of the epoxy resin [26]. The QDs dissolved in the epoxy were well protected from the moisture, exhibiting better moisture stability than that of the QSF structure. In conclusion, the composite particles with good optical and moisture stability can be applied in the fabrication of high-performance QD-based LED devices, which can be directly used in the component package processes.

## 4. Conclusions

In summary, we fabricated a flexible, optical QD-based luminescent film with an improved moisture performance by packaging QD/epoxy composite particles (QE CPs) into silicone resin. The optical behaviors of the CPF films were controlled by varying the concentration of QE CPs, thus achieving different lighting effects. As the QD concentration in QE CPs increased from 0.09 to 0.37 wt%, the diffuse transmittance of CPF films decreases gradually together with the enhancement of scattering ability. The absorptance and PL increased rapidly, which contributes to the change in radiant power and luminous flux. The radiant power decreased rapidly from 534.8 to 275.5 mW because of the blue light consumption, while the luminous flux increased from 86.9 to 110.0 lm. Compared to the QSF structure under similar CCT, the CPF structure achieved a 34.2% enhancement in luminous efficiency due to the increase in light excitation and improvement in light extraction. The refractive index mismatch between the epoxy and silicone resin led to the scattering of light to improve the QD excitation and light extraction performance. The CCE of the CPF structure was 29.8%, which was 32% higher than that of a QSF structure. Furthermore, the QE CPs contributed to preventing the composite particles from water; thus, the CPF films demonstrated higher moisture stability than the QS film, exhibiting a 53.9% improvement under 85 °C and 85% RH ambient conditions after aging for 118 h. We believe that QD/epoxy composite particles can be used for preparing flexible light-converting films with high optical and moisture performance.

## Figures and Tables

**Figure 1 nanomaterials-11-02100-f001:**
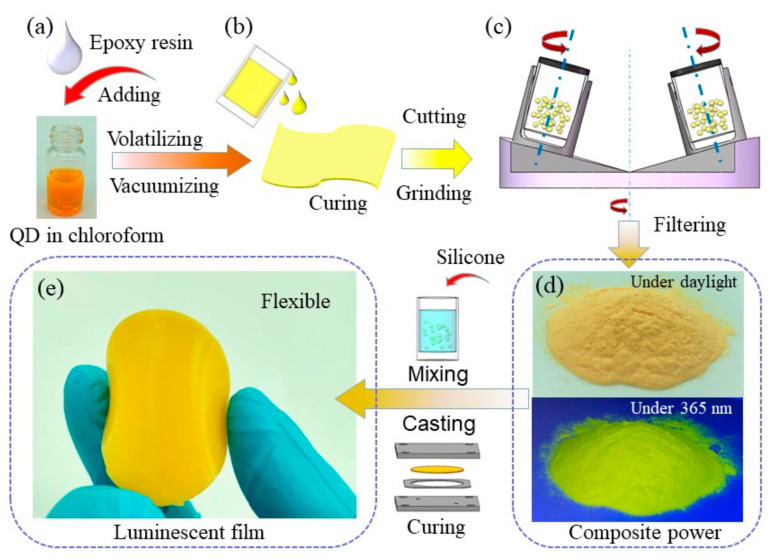
(**a**–**c**) Fabrication process of QD/epoxy composite particles. (**d**) Images of composite particles under daylight and 365 nm. (**e**) Images of flexible luminescent film fabricated by mold casting method.

**Figure 2 nanomaterials-11-02100-f002:**
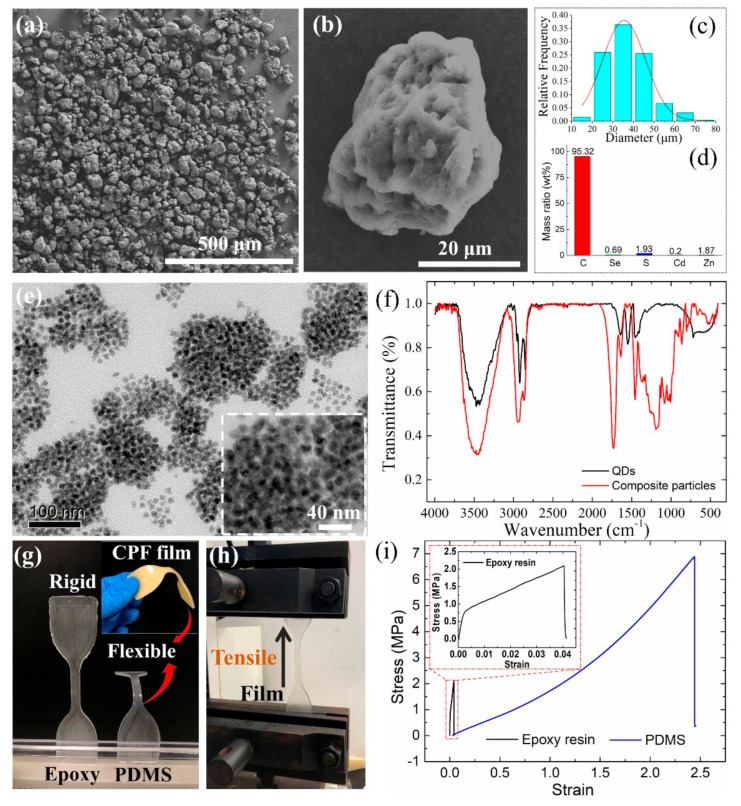
(**a**) SEM of composite particles. (**b**) Element mapping images of the composite particle. (**c**) Diameter distribution. (**d**) Element mass ratio distribution of corresponding mapping particle. (**e**) HRTEM images of QDs in epoxy resin. (**f**) FTIR spectra of QDs and composite particles. (**g**) Flexibility of PDMS film and rigidity of epoxy film. The inset shows the flexibility exhibition of CPF film. (**h**) Stress-strain curves measurement process. (**i**) Stress-strain curves of PDMS and epoxy films obtained from tensile test.

**Figure 3 nanomaterials-11-02100-f003:**
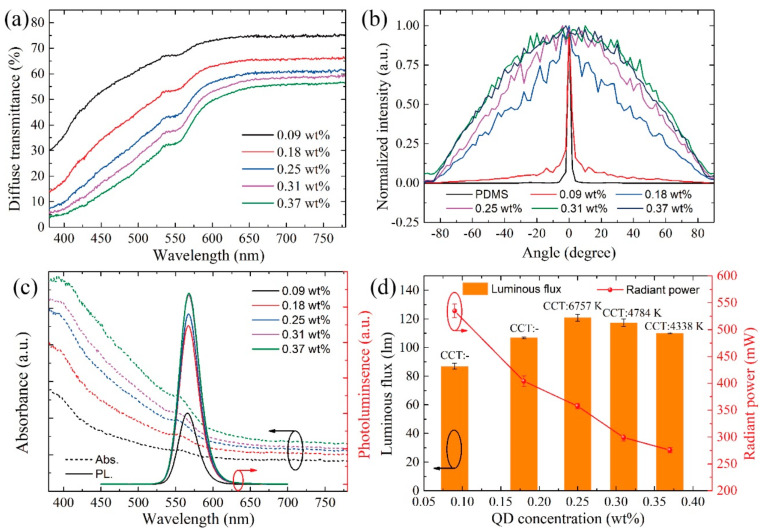
Optical properties of CPF films corresponding to various concentrations that varies from 0.09 to 0.37 wt%. (**a**) Diffuse transmittance. (**b**) BTDF. (**c**) Absorbance and PL. (**d**) Luminous flux and radiant power of LED devices with CPF films.

**Figure 4 nanomaterials-11-02100-f004:**
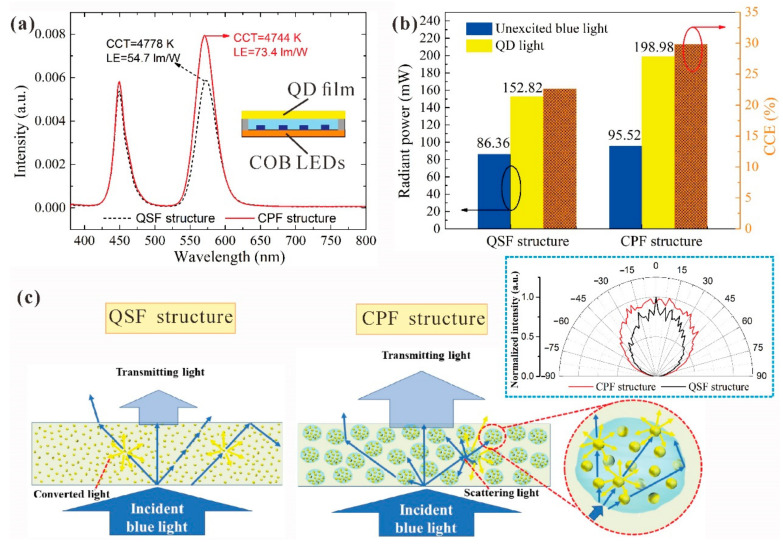
(**a**) Spectrum of devices with CPF structure and QSF structure under similar CCT at a driving current of 150 mA. The inset is the illustration of COB-LED device structure. (**b**) Radiant power of blue and QD light. (**c**) Schematics of light conversion and extraction for different structural films. The inset is the BTDF of two types of films under the same lighting effect.

**Figure 5 nanomaterials-11-02100-f005:**
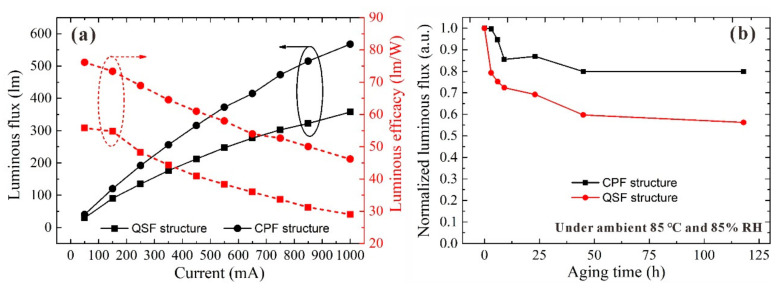
(**a**) Luminous flux and efficiency of LED devices with two kinds of structural films. (**b**) Normalized luminous flux during the aging process under 85 °C and 85% RH ambient.

## Data Availability

Data is contained within the article.

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
