# Peer review of "Optical Performance and Moisture Stability Enhancement of Flexible Luminescent Films Based on Quantum-Dot/Epoxy Composite Particles"

_nanomaterials, 2021, doi:10.3390/nano11082100_

Round 1

Reviewer 1 Report

The authors present a paper “Optical Performance and Moisture Stability Enhancement of Flexible Luminescent Films Based on Quantum-dot/Epoxy Composite Particles”. It is aimed to study the effects of the embedding of quantum dots into epoxy resin which is subsequently packed into a silica resin film. The article contains the usual set of experimental techniques for optical and photoluminescent properties examination, as well the mechanical test and moisture stability tests. Following the logic of the research, all physical characteristics discussed in the work are compared to a reference sample of quantum dots/ PDMS film meanwhile the discussion of the different studies in the literature available (10.1080/14328917.2019.1636175) could improve such paper with the focus on the synthesis procedure of a new type of composite materials based on the of QDs and two types of resins with the different refractive indices. The novelty of the technological procedure of the synthesis of ternary composite material corresponds to the scope of the MDPI Nanomaterials and the manuscript may be accepted for publication after some minor revision is fulfilled.

  1. Which kind of material are the QDs composed of? It is recommended to add the information on the material into the paper not referring to the previous work [23]. It makes easier the reading process when one realizes the composition of the composite and may suppose the compound used.
  2. What is the spatial distribution of the QD/epoxy composite particles in the PDMS films and its turn how the QDs are distributed in IK0010 epoxy resin?
  3. Is the change in curing parameters crucial for the spatial distribution and size of QD/epoxy resing particles?
  4. Was the refractive index mismatch between epoxy and silicon resin chosen intentionally? Could one expect a larger trapping effect for the blue light if another main matrix material was chosen?
  5. No experimental error estimation is given on the curves and values in the current. It is recommended to add this data to add scientific rigorous and it may serve for precise comparison to the data on a similar system described in the literature available.

The article requires a check of the language, formatting, and the use of the articles. Some examples are given below:

Lines 16-18: The photoluminescence and light extraction were improved due to the scattering properties of the composite particles, causing by the refractive index mismatch between the epoxy and silicone resin. --> … caused by …

Line 157 which may cause the change in incident light transfer direction --> which may cause a change in the incident light transfer direction

Line 170: stress–strain curve --> stress-strain curve

Line 184: Composite particles flexible films --> Composite particle flexible films

Author Response

Dear reviewer:

Thank you very much for your  comments concerning our manuscript with the ID number of nanomaterials-1311775, entitled “Optical Performance and Moisture Stability Enhancement of Flexible Luminescent Films Based on Quantum-dot/Epoxy Composite Particles”. Those are professional and valuable comments which have greatly helped us to revise and improve our research. According to the valuable comments, we have answered the reviewers’ questions and comments point by point. The amendments have been highlighted with changes underlined through the whole manuscript and we have revised the relevant part and highlighted with red line in the manuscript. We have gone through the manuscript carefully and repeatedly.

Reviewer 2 Report

In this manuscript, the author reports, ‘Optical Performance and Moisture Stability Enhancement of Flexible Luminescent Films Based on Quantum-dot/Epoxy Composite Particles’. The current study is on a topic of relevance and general interest to readers in this area. The manuscript is written with careful experiments and analysis. The authors should address the following questions before getting a possible publication.

Recommendation: Minor revisions needed as noted.

  1. In the 2.A. Composite particles preparation section, the author stated that ‘. Subsequently, the mixture was subjected to centrifugal stirring for 2 h to completely volatile the chloroform’. Here the author should mention the centrifugal speed in rpm.
  2. What is the quantum yield of the QDs?
  3. The formatting and grammatical errors in the article need to be checked carefully.
  4. It will be better if the author can include FTIR of the QDs, composite particles and luminescent films.
  5. The author should write the purpose for each test in one/two sentences (in brief) before explaining the results of the characterization techniques. Therefore, the logic and organization of this part will be enhanced
  6. The authors have cited relevant references in the Introduction section; however there are few that need to be included: Nanotechnology, 21(49), 495704; ACS Applied Nano Materials, 3(12), 11777-11790; Functional Composites and Structures, 1(2), 022001 (https://doi.org/10.1088/2631-6331/ab0c80); Nanomaterials, 9(8), 1100; Research on Chemical Intermediates 45.7 (2019): 3823-3853

Author Response

Dear reviewer:

Thank you very much for your comments concerning our manuscript with the ID number of nanomaterials-1311775, entitled “Optical Performance and Moisture Stability Enhancement of Flexible Luminescent Films Based on Quantum-dot/Epoxy Composite Particles”. Those are professional and valuable comments which have greatly helped us to revise and improve our research.

According to your valuable comments, we have carried out the related mearsurement and answered your questions and comments point by point.  We have gone through the manuscript carefully and repeatedly.

Please consider our response.
